# IRO/Shift Test Is Comparable to the Jobe Test for Detection of Supraspinatus Lesions

**DOI:** 10.3390/jpm12091422

**Published:** 2022-08-31

**Authors:** George Fieseler, René Schwesig, Julia Sendler, Jakob Cornelius, Stephan Schulze, Wolfgang Lehmann, Souhail Hermassi, Karl-Stefan Delank, Kevin Laudner

**Affiliations:** 1Clinic for Orthopedic and Trauma Surgery, Sports Medicine, Clinic Hannoversch Münden, 34346 Hannoversch Münden, Germany; 2Department of Orthopedic and Trauma Surgery, Martin-Luther-University Halle-Wittenberg, 06120 Halle, Germany; 3Clinic for Orthopedic, Trauma and Reconstructive Surgery, Georg August University Göttingen, 37075 Göttingen, Germany; 4Physical Education Department, College of Education, Qatar University, Doha 2713, Qatar; 5Department of Health Sciences, University of Colorado Colorado Springs, Colorado Springs, CO 80918, USA

**Keywords:** clinical, examination, rotator cuff, special test, sensitivity, specificity

## Abstract

To determine how the internal rotation and shift (IRO/shift) test compares to the gold standard of clinical tests (Jobe test) for diagnosing supraspinatus lesions and to confirm these clinical results with surgical findings, 100 symptomatic patients were clinically examined between October 2018 and November 2019. All 100 patients were evaluated using both the IRO/shift test and Jobe test. A total of 48 of these patients received surgical intervention. Based on these data, sensitivity, specificity, positive predictive value (PPV), negative predictive value (NPV), and accuracy for both the IRO/shift test and Jobe test were calculated. The IRO/shift test had a sensitivity of 96% (95% CI: 82–100%), specificity of 50% (95% CI: 27–73%), PPV of 73% (95% CI: 56–86%), NPV of 91% (95% CI: 59–100%), and an accuracy of 77% (95% CI: 63–88%). The Jobe test had a sensitivity of 89% (95% CI: 72–98%), specificity of 60% (95% CI: 36–81%), PPV of 76% (95% CI: 58–89%), NPV of 80% (95% CI: 52–96%), and an accuracy of 77% (95% CI: 54–81%). These results suggest that the IRO/shift test is comparable to the Jobe test, which is often viewed as the gold standard clinical examination for assessing supraspinatus lesions. This study was approved by the Ethics Commission of the Martin Luther University Halle-Wittenberg (reference number: 2018-05).

## 1. Introduction

The excessive mobility of the glenohumeral joint often comes at the expense of stability. The rotator cuff muscles are key stabilizers of the shoulder and are routinely stressed during activities of daily living, as well as during overhead sporting motions [1]. It is therefore not surprising that rotator cuff injuries account for the majority of all shoulder-related pathologies [2]. Research reported that 34% of the general population had symptomatic tears, while 65% had asymptomatic tears, and the prevalence of tears significantly increased with age beginning in the 50s [3]. This pathology was also noted as a common cause of morbidity among the elderly [1]. Not only is rotator cuff pathology common but it can also be very debilitating, with 40% of patients with a rotator cuff disorder complaining of persistent symptoms lasting longer than 12 months [4]. Common symptoms include pain, lost range of motion, weakness, and sleep disturbances [5,6]. This combination of symptoms contributes to the debilitating nature of this pathology and negatively impacts activities of daily living, performance in sport, and leisure activities, as well as an individual’s ability to adequately perform their job [5]. 

Due to the common occurrence and debilitating nature of rotator cuff injuries, it is imperative that such pathologies are accurately diagnosed to ensure appropriate treatment. There has been a plethora of studies investigating the efficiency of different clinical tests for rotator cuff pathology. Common highly favorable tests for supraspinatus lesions include the drop arm test, full can test, and Hawkin’s sign, just to name a few [7,8,9]. Although special tests such as the drop arm test have been shown to have good specificity (87–99%) and variable sensitivity (6–71%) [9,10], the Jobe’s test commonly outperforms the others with more consistent specificity (62–75%) and sensitivity (72–88%) [7,9,11] and is considered the gold standard of clinical testing [7,9]. 

Due to the wide ranges of specificity and sensitivity between tests, several reports have suggested that clinicians should not rely on just one test but instead use multiple tests to increase diagnostic accuracy [9,12,13,14]. As such, the lead author of this study recently developed the internal rotation and shift (IRO/shift) test to add to the existing special tests in an effort to enhance the diagnostic efficiency of supraspinatus tears. 

The objective of our study was to determine how the IRO/shift test compares to the gold standard of clinical tests (Jobe test) and confirm with surgical findings. More specifically, this study compared the specificity, sensitivity, positive predictive value (PPV), and negative predictive value (NPV) of the IRO/shift test and Jobe test compared to surgical findings. We hypothesized that the IRO/shift test would provide similar findings compared to those of the Jobe test and surgical findings and would be considered a beneficial clinical test to include in supraspinatus evaluation protocols.

## 2. Materials and Methods

### 2.1. Subjects

A sample of convenience was used among 100 patients who initially volunteered to participate in this study between October 2018 and November 2019. Subjects were 18 years or older and were patients at a single surgery and rehabilitation clinic that specializes in shoulder pathologies. All subjects provided informed consent prior to any data collection as approved by the Ethics Commission of the Martin Luther University Halle-Wittenberg (reference number: 2018-05). Patients were excluded from the study if they had a status of post instability, arthrosis, fracture, rheumatoid arthritis, or adhesive capsulitis or had upper extremity range of motion (ROM) restrictions. Among the 100 patients evaluated clinically, 48 were scheduled for arthroscopic surgery (31 male, 17 female, age 55.4 ± 11.4 years, height 1.76 ± 0.08 m, weight 85.2 ± 18.6 kg, and BMI 27.3 ± 4.5 kg/m²) between 2018 and 2019. Additional demographic information can be viewed in Table 1. Both partial and full-thickness tears identified during surgery were considered positive for a supraspinatus lesion and data analysis. Reasons for not receiving surgery among the remaining 52 patients included pre-operative and anesthetic contraindications, the patient not wanting to pursue surgical treatment, major and retractive supraspinatus lesion, being a declared candidate for primary reverse shoulder arthroplasty, or conservative treatment having been recommended. 

### 2.2. Procedures

A blinded, prospective research design was used for this study. Each of the 48 surgical patients was clinically evaluated using both the IRO/shift test and the Jobe test and subsequently received surgery. All pre-operative examinations were conducted by an experienced orthopedic physician who has been conducting shoulder surgeries since 2001 and who performs approximately 200 shoulder surgeries annually. 

For the IRO/shift test [15], each patient stood in a relaxed position with their feet shoulder-width apart. The patient was instructed to actively move their involved side arm behind their back and then internally rotate so that the dorsal aspect of their hand moved superiorly along their spine. At the end range of pain-free motion, the examiner then applied an additional passive movement of the arm into greater internal rotation and therefore elevated the hand further superiorly along the spine (Figure 1). Passively moving the shoulder into end-range adduction and internal rotation causes further anterior translation of the humeral head and subsequent increased pressure on the supraspinatus and any potential muscle defect, which results in increased pain (positive test). The intention of the IRO/shift test is to determine the presence of supraspinatus pathology; however, pain during this motion may also be caused by damage to the long head of the biceps. Therefore, biceps involvement must subsequently be ruled out. For this, the examiner performed subsequent special tests, such as the O’Brien’s, Speed’s, and/or Yergason’s tests. If pain was present during these additional special tests, indicating biceps involvement, then the IRO/shift test was considered negative; however, if the biceps special tests were negative, then the IRO/shift test was considered positive for supraspinatus pathology. 

### 2.3. Statistical Analysis

SPSS version 28.0 for Windows (IBM, Armonk, NY, USA) was used for all statistical analyses. Surgical findings were used to determine the sensitivity, specificity, positive predictive value (PPV), negative predictive value (NPV), and accuracy for both the Jobe test and IRO/shift test. Sensitivity of these diagnostic tests was defined as the test’s ability to demonstrate a positive test when the condition was actually present (i.e., true positive). Conversely, specificity was the test’s ability to produce a negative test when the pathology was actually absent (i.e., true negative). In addition to sensitivity and specificity, values for prediction were calculated to provide perspectives into the feasibility of the tests. The PPV determines the likelihood that a patient who presents with a positive clinical test has the pathology, while NPV determines the likelihood that a patient who presents with a negative test is actually free from that pathology. The accuracy of a test considers both sensitivity and specificity and thereby describes how correct the test is in identifying and excluding a pathology. 

A sample size of 70 was calculated based on the prevalence of surgical intervention of 50% (48/100) and an expected sensitivity of 90% (95% CI, width of 10%) [16].

## 3. Results

The IRO/shift test has shown good–excellent reliability (intrarater (ICC = 0.73, 95% CI: 60–82), interrater (ICC = 0.89, 95% CI: 81–94)), as well as 92% sensitivity (95% CI: 86–99%) and 67% specificity (95% CI: 50–84%) when confirmed with MRI. This test also demonstrated 86% positive predictive value and 80% negative predictive value [15].

### 3.1. Surgical Findings

Among the 48 patients who underwent surgery, 28 were found to have a supraspinatus tear (partial or full). Various pathologies were found among the 20 patients who did not have a supraspinatus tear including biceps tendon pathology, SLAP lesions, osteophytosis, bursitis, acromioclavicular joint arthrosis, and calcific tendinitis. Table 2 depicts the comparison of the IRO/shift test and the Jobe test with the surgical findings. 

### 3.2. Comparison of IRO/Shift Test and Jobe Test

Among the 28 patients with a supraspinatus lesion identified upon surgery, 26 (93%) presented with a positive IRO/shift test and 25 (89%) had a positive Jobe test. Among the 20 patients who had no evidence of supraspinatus lesion found during surgery, 10 (50%) had a negative IRO/shift test, while 12 (60%) had a negative Jobe test. For the IRO/shift test and Jobe test, a similar performance was calculated (Table 2). 

## 4. Discussion

Rotator cuff pathology is one of the leading orthopedic injuries with an increased prevalence of both partial and full-thickness tears as individuals age [17,18,19]. What makes this pathology even more difficult is that reliance on imaging (MRI, diagnostic ultrasound) alone can be detrimental due to the increased risk of inaccurate diagnoses [20,21,22]. As such, the purpose of this study was to compare the accuracy of the IRO/shift and Jobe tests with surgical intervention as opposed to MRI. Therefore, multiple clinical tests should be used during evaluation for enhanced diagnostic accuracy. Although numerous special tests have been investigated in the literature, there is still a paucity of information on which are most effective. The results of this study indicated that the IRO/shift test is a valid clinical test that is comparable to the Jobe test and should be considered when examining patients with potential supraspinatus pathology. 

We chose to compare the IRO/shift test with the Jobe test because the Jobe test is arguably the gold standard of clinical exams for evaluating superior supraspinatus pathologies. Jain et al. [23] used a predictive model to determine the probability of supraspinatus tears without using imaging. These predictors included sex, the involved-to-uninvolved ratio of external rotation strength, and results of the Jobe test and lift-off test. The authors reported that this model could accurately recognize a supraspinatus tear with satisfactory discrimination. However, this is not the first study to recognize the efficiency of the Jobe test. Numerous investigations have noted the Jobe test as being highly sensitive and specific for supraspinatus tears; however, there has been some disagreement on whether the test is more sensitive or specific [7,9,24,25]. Other studies have used surgical findings to determine the accuracy of other special tests compared to the Jobe test for diagnosing supraspinatus tears. Franca et al. [26] reported that compared to the Jobe test, the drop arm test was less sensitive but more specific. Similarly, the drop arm test had higher NPV but lower PPV. Zou et al. [27] found that the Jobe test and the hug-up test had similar accuracy, while the drop arm had less sensitivity, specificity, PPV, and NPV and the Hawkins test had similar specificity and PPV but less sensitivity and NPV than Jobe. Our study found that when confirmed with surgical findings, the Jobe test demonstrated good sensitivity (89%) and less specificity (60%); however, both PPV (76%) and NPV (80%) were laudable.

We believe that these results demonstrate the value of the IRO/shift test. However, past investigations have consistently shown that multiple clinical tests should be used because no single clinical test can sufficiently diagnose supraspinatus tears [9,12,13,14]. Furthermore, the use of multiple reliable and valid clinical tests may also decrease the clinicians need to rely on expensive imaging [7,9]. As such, we suggest clinicians incorporate the IRO/shift test into their existing protocol for clinical examination of supraspinatus pathology. 

There are a few limitations to this study worth noting. Patients must also have ample active and passive glenohumeral internal rotation and adduction ROM to perform the IRO/shift test. As such, this special test would not be appropriate for patients with severe restrictions or pain in these motions. Additionally, we chose to compare the IRO/shift test to the Jobe test because the Jobe test is predominantly considered to be the gold standard for clinical examinations of superior rotator cuff pathology; however, there are numerous other special tests that should be considered during evaluation, such as Hawkin’s sign, full can test, and drop arm test, to name but a few [7,8,9]. Future investigations are necessary to compare the clinical usefulness of the IRO/shift test with these other exams. Lastly, the involved number of patients did not meet our calculated sample size (48 vs. 70). Therefore, the results should not be overinterpreted.

## 5. Conclusions

The IRO/shift test had similar sensitivity and specificity to that of the Jobe test and demonstrated good validity for assessing supraspinatus pathology. As such, we would suggest clinicians consider using the IRO/shift test along with other clinical tests and advanced imaging preferred by the respective clinician.

## Figures and Tables

**Figure 1 jpm-12-01422-f001:**
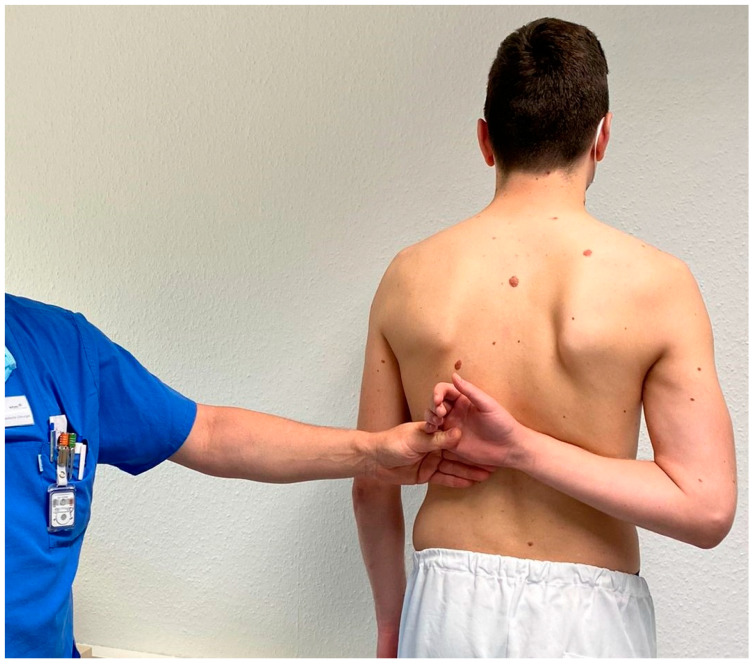
Passively applying elevation overpressure (shoulder adduction and internal rotation) to the patient’s end range of motion.

**Table 1 jpm-12-01422-t001:** Patient anthropometric and demographic data (*n* = 48).

Age (years)	55.4 ± 11.4
Height (m)	1.76 ± 0.08
Weight (kg)	85.2 ± 18.6
Body mass index (kg/m^2^)	27.3 ± 4.53
Sex (male: female)	31:17
Duration of pain (<6 months: >6 months)	2:46
accident anamnesis (yes: no)	11:37
pain localization (left: right)	21:27
Cause of the lesion (traumatic: after shoulder dislocation: degenerative/impingement)	9:2:37

Reported as mean ± standard deviation.

**Table 2 jpm-12-01422-t002:** IRO/shift and Jobe Tests compared to surgical findings.

Test	IRO/Shift Test % (95% CI)	Jobe Test % (95% CI)
Sensitivity Specificity PPV NPV Accuracy	96 (82–100) 50 (27–73) 73 (56–86) 91 (59–100) 77 (63–88)	89 (72–98) 60 (36–81) 76 (58–89) 80 (52–96) 77 (54–81)

CI: confidence interval; PPV: positive predictive value; NPV: negative predictive value.

## Data Availability

Please contact the corresponding author for inquiries regarding study data.

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
