# Peer review of "IRO/Shift Test Is Comparable to the Jobe Test for Detection of Supraspinatus Lesions"

_jpm, 2022, doi:10.3390/jpm12091422_

Round 1
Reviewer 1 Report
Title:
Ok, although my recommendation is to write a title with the main finding, for example
“…. IRO/shift test is comparable to the Jobe test….”
Abstract:
Ok.
Introduction:
Line 32: “Research has reported that 34% of the general population have
symptomatic tears, while 65% have asymptomatic tears [3].”
I would edit this sentence and divide by age or add a sentence about the prevalence at different ages
Line 45: edit this sentence regarding the various tests
Neer’s sign – This test is more suitable for impingement than SSP lesion
external rotation lag sign – more for ISP
Line 47: address the drop test which is the most specific for full thickness RC tears (specificity ~98%)
Please add a paragraph regarding the IRO\shift test
What is the rationale behind this test?
Did you develop it or was it described before?
Methods:
Line 64: The authors stated that “Patients were excluded from the study if they were status post glenohumeral dislocation, instability….”
In table 1 which elaborate the demographic and anthropometric characteristics of the 42 patients that were included in the study and underwent surgical treatment, there are 2 patients that the cause of lesion is after shoulder dislocation… Please clarify the inclusion\exclusion criteria.
Line 81: was the evaluator was blinded to the purpose of the study? Was the evaluator being independent or one of the authors? Please clarify.
Line 84: please elaborate the rationale and anatomy for the IRO\shift test
It is known that the dominant hand has less internal rotation than the non-dominant hand, did you evaluate this issue? Did you collect this data?
If so, elaborate , as this may affect the results and accuracy of the test
If not – address in the limitations.
Line 93 – for testing a biceps pathology, please explain why the authors chose the O’Brien’s test and not Speed\Yergasons’ tests?
Line 97: “The IRO/shift test has demonstrated good-to-excellent reliability (intrarater (ICC=0.73, 95% CI: 60-82), interrater (ICC=0.89, 95% CI: 81-94)), as well as 92% sensitivity (95% CI: 86-99%) and 67% specificity (95% CI: 50-84%) when confirmed upon MRI. Predictive values were also high with an 86% positive predictive value and 80% negative predictive value [10].”
This paragraph is more suitable for results section.
Line 102: was sample size calculated prior to the study? For optimal calculation of the predictors?
Results:
Please clarify or elaborate what was the added value of surgical findings if a comparison to MRI findings was performed? Where there any cases that surgery revealed a SSP tear that wasn’t demonstrated in the pre op MRI?
Line 130 – a repetition for table 2. Please delete and refer to the table.
Discussion:
Please address other studies that compared different test to Jobe’s test.
If the IRO\shift has been described before, please refer to the studies that tested its effectiveness\. accuracy. If the test was described for the first time in this study, see my comment in the methods section and detail the rationale behind this test. This is for the benefit of the readers, because as the text is now written, it is not clear.
Line 159: “Upon comparison of the IRO/shift test with the Jobes’ test we found that these clinical
tests performed similarly. The IRO/shift test demonstrated higher sensitivity (96%) compared to the Jobe test (89%) but had less specificity (50% compared to 60%). For PPV, the IRO/shift and Jobe tests performed similarly (73% and 76% respectively), while the IRO/shift test had a higher NPV (91%) compared to Jobe (80%).”
This is a repetition of the results. Please remove.
Limitations:
See my comment regarding dominant hand in the method section.
Conclusions:
Line 183: “strongly recommend clinicians consider” please edit this sentence
Use “may” or “suggest” instead of “strongly recommend”
Figures: For the sake of the readers, I suggest adding a figure or a video to demonstrate the IRO\shift test.
Author Response
TITLE
Point 1: Ok, although my recommendation is to write a title with the main finding, for example
“…. IRO/shift test is comparable to the Jobe test….”
Response 1:
Thank you for this feedback and suggestion. We changed the title as follows:
@ line 2-3: IRO/shift test is comparable to the Jobe test for detection of Supraspinatus Lesions
INTRODUCTION
Point 2: Line 32: “Research has reported that 34% of the general population have symptomatic tears, while 65% have asymptomatic tears [3].” I would edit this sentence and divide by age or add a sentence about the prevalence at different ages.
Response 2:
Thank you for this feedback. We edit this sentence/ paragraph as suggested:
@ line xx-yy: Research has reported that 34% of the general population have symptomatic tears, while 65% have asymptomatic tears and the prevalence of tears significantly increases with age beginning in the 50s [3].
Point 3: Line 45: edit this sentence regarding the various tests Neer’s sign – This test is more suitable for impingement than SSP lesion external rotation lag sign – more for ISP.
Response 3:
We changed this sentence as follows:
@ line xx-yy: Common highly favorable tests for supraspinatus lesions include the drop arm test, full can test, Neer’s sign, external rotation lag sign, and Hawkin’s sign, just to name a few [7-9].
Point 4: Line 47: address the drop test which is the most specific for full thickness RC tears (specificity ~98%) Please add a paragraph regarding the IRO\shift test What is the rationale behind this test? Did you develop it or was it described before?
Response 4:
We changed this sentence as follows:
@ line xx-yy: Although, these special tests such as the drop arm test have been shown to have good sensitivity and specificity (87-99%), but variable sensitivity (.06-71%) [9,10], the Jobe’s test commonly outperforms these others with more consistent specificity (62-75%) and sensitivity (72-88%) [7,9,11]and is considered the gold standard of clinical testing [7,9].
Due to the wide ranges of specificity and sensitivity between tests several reports have suggested clinicians don’t rely on just one test, but instead use multiple tests to increase diagnostic accuracy [9,12-14]. As such, the lead author of this study recently developed the internal rotation and shift (IRO/shift) test to add to the existing special tests in an effort to enhance the diagnostic efficiency of supraspinatus tears.
METHODS
Point 5: Line 64: The authors stated that “Patients were excluded from the study if they were status post glenohumeral dislocation, instability….”. In table 1 which elaborate the demographic and anthropometric characteristics of the 42 patients that were included in
the study and underwent surgical treatment, there are 2 patients that the cause of lesion is after shoulder dislocation…Please clarify the inclusion\exclusion criteria.
Response 5:
Authors response: Thank you for catching this error and we apologize for the confusion. Dislocation should not have been listed as an exclusion factor. We have deleted this from the “subjects” section.
Point 6: Line 81: was the evaluator was blinded to the purpose of the study? Was the evaluator being independent or one of the authors? Please clarify.
Response 6:
Authors response: the evaluator was not blinded to the purpose of the study, but he was blind to any other test results (ultrasound, x-rays, MRI). Therefore, the examiner had no knowledge if a patient was suffering from a RC pathology or any subsequent shoulder pathology.
Point 7: Line 84: please elaborate the rationale and anatomy for the IRO\shift test.
Response 7:
We described the rationale and anatomy for the IRO\shift test in more detail as suggested:
@ line xx-yy: Passively moving the shoulder into end range adduction and internal rotation causes further anterior translation of the humeral head and subsequent increased pressure on the supraspinatus and any potential muscle defect, which results in increased pain (positive test).
Point 8: Line 84: It is known that the dominant hand has less internal rotation than the non-dominant hand, did you evaluate this issue? Did you collect this data? If so, elaborate, as this may affect the results and accuracy of the test. If not – address in the limitations.
Response 8:
Authors response: hand dominance is not a factor with the IRO/Shift test. The clinician is simply applying passive pressure to the end range of adduction and internal rotation. The amount of passive motion available is respective of the total range of motion in that shoulder, regardless of if it is the dominant or non-dominant. For example, a shoulder with less ROM, like the dominant arm, would have less passive motion available, but it would still be moved to its end range and cause just as much pressure as if applied to a non-dominant shoulder that had more range of motion, because that dominant shoulder would still be moved to its respective end range.
Point 9: For testing a biceps pathology, please explain why the authors chose the O’Brien’s test and not Speed\Yergasons’ tests?
Response 9:
Authors response: similar to our suggestion of using multiple tests to increase diagnostic accuracy of detecting supraspinatus tears, we also used multiple tests to determine biceps pathology. We added the explanation as follows:
@ line xx-yy: For this, the examiner performed subsequent special tests, such as the O’Brien’s, Speed’s, and/or Yergason’s tests.
Point 10: Line 97: “The IRO/shift test has demonstrated good-to-excellent reliability (intrarater (ICC=0.73, 95% CI: 60-82), interrater (ICC=0.89, 95% CI: 81-94)), as well as 92% sensitivity (95% CI: 86-99%) and 67% specificity (95% CI: 50-84%) when confirmed upon MRI. Predictive values were also high with an 86% positive predictive value and 80% negative predictive value [10].” This paragraph is more suitable for results section.
Response 10:
We moved this paragraph to the results section as suggested:
@ line 114-118: The IRO/shift test has demonstrated good-to-excellent reliability (intrarater (ICC=0.73, 95% CI: 60-82), interrater (ICC=0.89, 95% CI: 81-94)), as well as 92% sensitivity (95% CI: 86-99%) and 67% specificity (95% CI: 50-84%) when confirmed upon MRI. Predictive values were also high with an 86% positive predictive value and 80% negative predictive value [10].
Point 11: Line 102: was sample size calculated prior to the study? For optimal calculation of the predictors?
Response 11:
@ line 135-137: The sample size was calculated based on Buderer (1996). Based on a prevalence of surgical intervention of 50% (48/100) and an expected sensitivity of 90% (95%CI, width of 10%) a sample size of 70 patients is necessary.
Furthermore, we added the following sentence in the limitations:
@ line 207-209: A sample size of 70 was calculated based on the prevalence of surgical intervention of 50% (48/100) and an expected sensitivity of 90% (95% CI, width of 10%) [16].
Buderer, N. M. F. "Statistical methodology: I. Incorporating the prevalence of disease into the sample size calculation for sensitivity and specificity." Acad Emerg. Med. 1996;3:895-900.
RESULTS
Point 12: Please clarify or elaborate what was the added value of surgical findings if a comparison to MRI findings was performed? Where there any cases that surgery revealed a SSP tear that wasn’t demonstrated in the pre op MRI?
Response 12:
Thank you for this valuable comment. We added the following information in this context.
@ line xx-yy: What makes this pathology even more difficult is that reliance on imaging (MRI, diagnostic ultrasound) alone can be detrimental due to increased risk of inaccurate diagnoses [19-21]. As such, the purpose of this study was to compare the accuracy of the IRO/Shift and Jobe tests with surgical intervention as opposed to MRI.
Point 13: Line 130 – a repetition for table 2. Please delete and refer to the table.
Response 13:
Thank you for this comment. We changed this paragraph to avoid a repetition for table 2 as follows:
@ line 132-138: For the IRO/shift test and Jobe test, a similar performance was calculated (Table 2). this resulted in a sensitivity of 96% (95% CI: 82-100%), specificity of 50% (95% CI: 27-73%), PPV of 73% (95% CI: 56-86%), NPV of 91% (95% CI: 59-100%), and an accuracy of 77% (95% CI: 63-88%). For the Jobe test (Table 2), this resulted in a sensitivity of 89% (95% CI: 72-98%), specificity of 60% (95% CI: 36-81%), PPV of 76% (95% CI: 58-89%), NPV of 80% (95% CI: 52-96%), and an accuracy of 77% (95% CI: 54-81%).
DISCUSSION
Point 14: Please address other studies that compared different test to Jobe’s test.
Response 14:
We searched and added other studies for a better comparison with the Jobe test.
@ line xx-yy: Other studies have used surgical findings to determine the accuracy of other special tests compared to the Jobe test for diagnosing supraspinatus tears. Franca et al [25]., reported that compared to the Jobe test, the drop arm test was less sensitive, but more specific. Similarly, the drop arm test had higher NPV but lower PPV. Zou et al [26]., found that the Jobe test and the Hug-up test had similar accuracy, while the drop arm had less sensitivity, specificity, PPV, and NPV and the Hawkins test had similar specificity and PPV, but less sensitivity and NPV than Jobe.
Point 15: If the IRO\shift has been described before, please refer to the studies that tested its effectiveness\. accuracy. If the test was described for the first time in this study, see my comment in the methods section and detail the rationale behind this test. This is for the benefit of the readers, because as the text is now written, it is not clear.
Response 15:
We improved it as suggested in the previous comment.
@ line xx-yy: …
Point 16: Line 159: “Upon comparison of the IRO/shift test with the Jobes’ test we found that these clinical tests performed similarly. The IRO/shift test demonstrated higher sensitivity (96%) compared to the Jobe test (89%) but had less specificity (50% compared to 60%). For PPV, the IRO/shift and Jobe tests performed similarly (73% and 76% respectively), while the IRO/shift test had a higher NPV (91%) compared to Jobe (80%).” This is a repetition of the results. Please remove.
Response 16:
We removed this repetition as suggested.
@ line 161-165: Upon comparison of the IRO/shift test with the Jobe test we found that these clinical tests performed similarly. The IRO/shift test demonstrated higher sensitivity (96%) com-pared to the Jobe test (89%) but had less specificity (50% compared to 60%). For PPV, the IRO/shift and Jobe tests performed similarly (73% and 76% respectively), while the IRO/shift test had a higher NPV (91%) compared to Jobe (80%).
LIMITATIONS
Point 17: See my comment regarding dominant hand in the method section.
Response 17:
Please see previous authors’ description of why dominance and subsequent variations in ROM are irrelevant to the IRO/Shift test.
CONCLUSION
Point 18: Line 183: “strongly recommend clinicians consider” please edit this sentence
Use “may” or “suggest” instead of “strongly recommend”.
Response 18:
Thanks a lot for this advise. We improved the sentence as suggested:
@ line 184-186: As such, we would suggest clinicians consider using the IRO/shift test along with other clinical tests and advanced imaging preferred by the respective clinician.
FIGURES
Point 19: For the sake of the readers, I suggest adding a figure or a video to demonstrate the IRO\shift test.
Response 19:
In line with your suggestion, we created and added a new figure for a better understanding of the IRO/shift test in the methods (figure 1).
Reviewer 2 Report
The paper Comparison of the Internal Rotation and Shift Test with the 2 Jobe Test for Detection of Supraspinatus Lesions is very interesting, written in a concise way, easy to read. In practice, the diagnosis of rotator cuff lesions is very difficult and the use of well-proven clinical tests for damage, e.g. the supraspinatus is clinically significant. However, the paper contains additional information on additional radiological tests, such as ultrasound or MRI. It seems that they would be helpful especially that in about 40% of patients no supraspinatus lesion was found ( 20 patients from 48)Author Response
The paper Comparison of the Internal Rotation and Shift Test with the 2 Jobe Test for Detection of Supraspinatus Lesions is very interesting, written in a concise way, easy to read. In practice, the diagnosis of rotator cuff lesions is very difficult and the use of well-proven clinical tests for damage, e.g. the supraspinatus is clinically significant. However, the paper contains additional information on additional radiological tests, such as ultrasound or MRI. It seems that they would be helpful especially that in about 40% of patients no supraspinatus lesion was found ( 20 patients from 48)
Authors response: the MRI data was presented in a previous study we conducted. The reference for this work has now been added for clarification.
Added reference: The IRO/shift test has shown good-excellent reliability (intrarater (ICC=0.73, 95% CI: 60-82), interrater (ICC=0.89, 95% CI: 81-94)), as well as 92% sensitivity (95% CI: 86-99%) and 67% specificity (95% CI: 50-84%) when confirmed upon MRI. This test also showed demonstrated 86% positive predictive value and 80% negative predictive value [15].
Round 2
Reviewer 1 Report
The initial concerns in the manuscript have been addressed.
Reviewer 2 Report
The manuscript has been sufficiently improved